# Genome-Wide Identification of the AP2/ERF Gene Family and Functional Analysis of *PgAP2/ERF187* Under Cold Stress in *Panax ginseng* C. A. Meyer

**DOI:** 10.3390/plants14182922

**Published:** 2025-09-20

**Authors:** Yihan Wang, Shurui Wang, Xiangru Meng, Ping Wang, Hongmei Lin, Peng Di, Yingping Wang

**Affiliations:** State Local Joint Engineering Research Center of Ginseng Breeding and Application, Jilin Agricultural University, Changchun 130118, China; 20221860@mails.jlau.edu.cn (Y.W.); 20241645@mails.jlau.edu.cn (S.W.); mengxiangru0814@163.com (X.M.); ping@mails.jlau.edu.cn (P.W.); di@jlau.edu.cn (P.D.)

**Keywords:** AP2/ERF gene family, cold stress, ABA, *Panax ginseng*

## Abstract

*Panax ginseng* C. A. Meyer (*P. ginseng*) is a medicinal plant rich in bioactive components such as ginsenosides, polysaccharides, and volatile oils and is widely used in both the pharmaceutical and food industries. While the AP2/ERF gene family is well-documented to play crucial roles in plant growth, development, and defense responses, functional studies on this gene family in *P. ginseng* remain unreported. Our genome-wide analysis identified 318 PgAP2/ERF family members, which are classified into five subfamilies: AP2, DREB, ERF, RAV, and Soloist. Homology analysis revealed that segmental duplication serves as the primary evolutionary driver for the *PgAP2/ERF* gene family in *P. ginseng*. RT-qPCR analysis demonstrated that all *PgAP2/ERF* members in the DREB-A1 subgroup respond to cold stress. Specifically, we found that the DREB-A1 member *PgAP2/ERF187* plays a pivotal role in the cold stress response, with its expression specifically induced by ABA. Overexpression of *PgAP2/ERF187* in Arabidopsis significantly enhanced the expression of cold tolerance-related genes. Subcellular localization analysis confirmed the co-localization of PgABF and PgAP2/ERF187 in the nucleus. Combining transcription factor interaction predictions and yeast one-hybrid experiments, we propose that PgABF likely regulates *PgAP2/ERF187* expression by directly binding to its promoter region. These findings unveil the potential mechanism of the “PgABF-PgAP2/ERF187” regulatory module within the ABA signaling pathway during *P. ginseng*’s cold stress adaptation, thereby providing novel theoretical insights into the molecular mechanisms underlying *P. ginseng*’s cold resistance.

## 1. Introduction

The APETALA2/Ethylene Responsive Factor (AP2/ERF) gene family is a plant-specific transcription factor family that plays extensive roles in plant growth and development, stress responses, and hormone signaling regulation [1]. Members of this family contain one or two AP2/ERF domains, each consisting of 60 to 70 conserved amino acid residues. These genes have been identified in various plant species, including *Arabidopsis thaliana* (*A. thaliana*) [1], *Oryza sativa* (*O. sativa*) [2], *Myrica rubra* [3], and *Medicago sativa* [4]. The AP2/ERF domain is characterized by one α-helix and three β-sheet regions and is classified into five subfamilies based on structural and functional features: AP2 (APETALA2), ERF (Ethylene-Responsive Factor), DREB (Dehydration-Responsive Element Binding Protein), RAV (Related to ABI3/VP1), and the Soloists subfamily [5]. The AP2 subfamily contains two AP2/ERF domains. In 1994, the first AP2 gene was isolated in *A. thaliana* and was shown to regulate flower initiation and development [6]. The RAV subfamily contains one AP2 domain and one B3 domain. Studies have indicated that RAV family members play important roles in regulating bud growth, leaf senescence, and responses to biotic and abiotic stresses [7]. The ERF and DREB subfamilies are the largest branches of the AP2/ERF superfamily [5], each containing a single AP2 domain. These transcription factors regulate downstream gene expression by binding to cis-acting elements such as the GCC-box or DRE/CRT via their AP2/ERF domain, thereby mediating plant responses to environmental changes and performing corresponding biological functions [8]. The GCC-box is a cis-element composed of the core sequence “GCCGCC” and is primarily located in the promoter regions of genes related to pathogen proteins. In contrast, the DRE/CRT element contains the core sequence “CCGAC” [9] and is primarily found in promoters of genes induced by abiotic stress [10].

Throughout their growth, plants frequently encounter numerous challenges, including pathogen infection and harsh environmental conditions such as low temperature [10], drought [11], and salinity [12]. Cold stress triggers a series of physiological and biochemical changes in plants, including membrane-lipid peroxidation caused by reactive oxygen species (ROS) accumulation [13], decreased photosynthetic efficiency [14], and disturbed osmotic balance [15]. To counteract these adverse effects, plants have evolved sophisticated adaptive strategies: enhancing the activities of antioxidant enzymes (e.g., superoxide dismutase and catalase) to scavenge excess ROS [13], accumulating osmolytes such as proline and soluble sugars to maintain cell turgor, and adjusting membrane-lipid composition to preserve membrane fluidity [14]. In recent years, researchers have discovered that the DREB (CBF) subfamily, a well-characterized group within the AP2/ERF gene family, serves as a key regulator specifically responding to cold stress. When CBF homologous genes from *O. sativa*, *Lolium perenne*, *Zea mays*, *Hordeum vulgare*, and *Triticum aestivum* are overexpressed in transgenic *Nicotiana tabacum* or *A. thaliana*, they significantly activate the expression of cold-induced genes in the CBF regulon, thereby enhancing cold tolerance [16]. CBF proteins specifically bind to the dehydration-responsive element DRE/CRT (A/GCCGAC) in the *RD29A* promoter to activate downstream gene expression. However, binding specificity may vary among CBF members; for example, Vitis vinifera CBF5 does not bind DRE/CRT, whereas ZjDREB1.4 shows a binding preference for GCCGAC over ACCGAC [17,18]. The induction level of CBF is influenced by genotypic variation and promoter polymorphisms, and its overexpression can mitigate cold damage by enhancing antioxidant enzyme activity and osmolyte accumulation [19]. Notably, CBF function is regulated at multiple levels by kinases such as BIN2 [20] and CRPK1 [21], and it forms a complex interaction network with the JA signaling pathway, DELLA proteins, and light signaling components like phyB/PIFs [22]. Additionally, exogenous ABA, circadian rhythms, and eugenol can also modulate CBF expression [23]. Other AP2/ERF members, such as *CRF2/3*, enhance cold adaptation by promoting lateral root development [24]. These findings provide critical theoretical insights into the molecular mechanisms underpinning plant cold stress responses.

The *Panax ginseng* C. A. Meyer (*P. ginseng*) is a perennial herbaceous plant belonging to the *Panax genus* of the *Araliaceae* family, has been used as a medicinal species in China for thousands of years. Its bioactive components (e.g., ginsenosides and polysaccharides) have been demonstrated to exhibit multi-target regulatory effects, showing significant clinical potential in improving cardiovascular function, regulating glucose metabolism [25], and enhancing neurocognition [26]. However, functional studies on the AP2/ERF gene family in *P. ginseng* have not yet been reported. To gain a deeper understanding of the functions of AP2/ERF transcription factors in *P. ginseng*, this study conducted systematic investigations, including genomic analysis, gene function prediction, and qRT-PCR transcriptome analysis, to screen for cold resistance-related genes. Using biotechnological approaches such as transgenic *A. thaliana* and yeast one-hybrid assays, we demonstrated the critical role of *PgAP2/ERF187* in cold stress resistance. These findings not only provide a foundation for elucidating the gene functions and molecular evolutionary mechanisms of AP2/ERF transcription factors in *P. ginseng* but also offer valuable insights for breeding superior *P. ginseng* varieties and improving germplasm through transgenic technology.

## 2. Results

### 2.1. Identification and Analysis of the AP2/ERF Gene Family in P. ginseng

The HMM profile (PF00847) of the AP2/ERF gene family was downloaded from the PFAM protein database. Using HMMER3 v3.3.2 software, protein sequences containing the AP2/ERF domain were identified, and conserved domains were further validated through NCBI and SMART. A total of 318 non-redundant AP2/ERF genes were ultimately confirmed. These genes were renamed PgAP2/ERF1 to PgAP2/ERF318 based on their chromosomal locations in *P. ginseng* (Appendix A). Protein characteristics of the *P. ginseng* AP2/ERF family members were analyzed. These proteins encode polypeptides ranging from 118 (PgAP2/ERF228) to 1738 (PgAP2/ERF131) amino acids, with molecular weights varying between 13.236 kDa and 19.8 kDa (Appendix A). Among them, 317 PgAP2/ERF proteins exhibited an average hydrophobicity (GRAVY) score below 0, while only one had a score above 0. The average isoelectric point (pI) was 6.89, indicating that most PgAP2/ERF proteins are near-neutral (Appendix A). Additionally, the predicted aliphatic index ranged from 43.59 to 101.6 (Appendix A). The instability index varied between 29.11 and 77.34, with 289 PgAP2/ERF proteins showing an instability index above 40, suggesting that these proteins are likely unstable and prone to denaturation or degradation (Appendix A). Subcellular localization predictions revealed that 83.33% of PgAP2/ERF proteins are localized in the nucleus, while a minority are targeted to chloroplasts, mitochondria, and other organelles (Appendix A).

### 2.2. Phylogenetic Analysis of PgAP2/ERF Proteins

To clarify the evolutionary relationships of *PgAP2/ERF* genes, a neighbor-joining (NJ) phylogenetic tree was constructed through multiple sequence alignment of 494 AP2/ERF protein sequences, including 318 AP2/ERF members in *P. ginseng* and 176 AP2/ERF *A. thaliana* members (Figure 1). The family was divided into five subfamilies: the Soloist subfamily with special functional domains contained only 4 members (including PgAP2/ERF85); the RAV subfamily containing both AP2/ERF domains and additional B3 DNA-binding domains comprised 11 members; and the AP2 subfamily with two AP2/ERF domains included 79 members. Among them, the ERF subfamily formed the largest branch with 251 members, followed by the DREB subfamily with 149 members (Figure 1). Due to their highly similar domain characteristics, ERF and DREB subfamilies were classified into the same major group but were further divided into DREB (Group A) and ERF (Group B) subfamilies based on domain differences, totaling 12 subgroups (A-1 to A-6 and B-1 to B-6) (Figure 1). Subgroup analysis showed that the B3 subgroup was the largest branch while the A3 subgroup was the smallest (Figure 1). AP2/ERF members within the same subgroup exhibited high conservation in gene structure and may possess similar biological functions.

### 2.3. Analysis of Conserved Motifs and Gene Structure of PgAP2/ERF Proteins

Analysis of conserved domains in PgAP2/ERF family proteins using MEME v5.5.5 software revealed that all members contain complete AP2 domains (Appendix A). Significant differences in domain composition were observed among subfamilies: AP2 subfamily members exhibited dual AP2 domain characteristics and specifically contained motif4, motif5 and motif9; all four RAV subfamily members contained B3 domains; in the DREB subfamily, A1, A4 and A5 subgroups all contained motif6, with the A1 subgroup additionally containing motif7 (Appendix A). Notably, PgAP2/ERF107, PgAP2/ERF131 and PgAP2/ERF223 from the DREB-A2 subgroup contained PHD SF, HepA and Laccase domains, respectively, while PgAP2/ERF85, the sole member of the Soloist subgroup, contained a PKc-like domain (Appendix A). Some ERF subgroup members (PgAP2/ERF43/85/110/127/264/286) possessed additional domains including SGNH hydrolase, Ribosomal L19e and PLN02872, besides the AP2 domain (Appendix A). These differences in domain combination patterns may reflect the molecular basis of functional specialization during evolution. Gene structure analysis showed that 85.22% of *PgAP2/ERF* members contain 1–2 exons, while 30.5% possess 1–6 UTR regions (Appendix A). AP2 subfamily members exhibited significantly more exons than other subfamilies, with 62.03% containing 6–10 exons (Appendix A), suggesting this complex structure may influence transcriptional efficiency and intracellular protein concentration.

### 2.4. Collinearity Analysis of the PgAP2/ERFGene Family

Analysis of chromosomal distribution revealed an uneven dispersion pattern of 318 *PgAP2/ERF* family members across 24 chromosomes (Figure 2). Significant variations were observed in gene numbers per chromosome, ranging from 8 to 20 (Figure 2a). Gene duplication analysis identified 413 homologous duplication events involving 285 *PgAP2/ERF* genes distributed across all 24 chromosomes (Appendix A). Chr11 and Chr14 displayed the highest duplication density (19 duplicated genes each), contrasting with Chr15’s minimal density (6 duplicated genes) (Figure 2a). Fragment duplications predominated (400 pairs, 96.85%), with only 13 tandem duplications observed (Figure 2a), confirming fragment duplication’s dominant role in family expansion. Selection pressure analysis revealed Ka/Ks ratios significantly below 1 for all duplicated gene pairs (Appendix A), indicating strong purifying selection that maintained functional conservation while suppressing functional divergence. Interspecies synteny analysis demonstrated evolutionary relationships between *PgAP2/ERF* genes and four related species (Figure 2b). *PgAP2/ERF* showed 3, 4, 2 and 7 homologous gene pairs with *A. thaliana*, *Panax quinquefolius*, *Panax stipuleanatus* and *Panax japonicus*, respectively (Appendix A). The highest synteny with *P. japonicus* (7 pairs) suggests strong evolutionary conservation, whereas minimal synteny with *P. stipuleanatus* (2 pairs) may reflect differential selection pressures or genomic reorganization events (Figure 2b).

### 2.5. Cis-Acting Element Analysis of PgAP2/ERF Family Gene Promoters

Analysis of cis-acting elements in the promoter regions (2000 bp upstream) of 318 *PgAP2/ERF* genes identified 6756 regulatory elements, which were classified into five categories: light response, hormone response, growth and development, MYB binding, and stress response (Figure 3a). The results showed that light-responsive elements accounted for the highest proportion (2649, 39.17%), followed by hormone-responsive elements (2485) (Figure 3b). Among hormone-responsive elements, abscisic acid (ABRE) and methyl jasmonate (CGTCA-motif/TGACG-motif) related elements constitute 35.7% (818) and 42.4% (967), respectively (Figure 3b). Additionally, 455 growth and development-related elements, 426 MYB binding elements, and 748 stress-responsive elements were identified, including 133 cold response elements (LTR) (Figure 3b). Given the multiple roles of abscisic acid (ABA) in plant growth and development, stress response, and hormonal regulation [27], these findings suggest that the PgAP2/ERF gene family may play a crucial role in the regulatory network of plant responses to environmental stresses, particularly cold stress, through the ABA signaling pathway.

### 2.6. Expression Patterns of PgAP2/ERF Family Genes Under Hormone and Stress Treatments

The AP2/ERF family plays crucial roles in plant stress responses. Transcriptome analysis of *P. ginseng* under cold, drought and salt stress treatments revealed that 71.43% of DREB subgroup members showed significant responses to salt and cold stresses. Notably, DREB-A1 subgroup members including *PgAP2/ERF148* were significantly upregulated under both cold and drought stresses, with 10 genes (including *PgAP2/ERF148*) specifically responding to cold stress (Figure 4a). Analysis of published transcriptome data for hormone responses indicated that 65.93% of DREB subgroup members were sensitive to GA and ABA treatments (Figure 4b). *PgAP2/ERF26*, *PgAP2/ERF102* and *PgAP2/ERF194* specifically responded to IAA treatment, while *PgAP2/ERF187* from DREB-A1 subgroup functioned as a specific regulator in ABA signaling pathway (Figure 4b).

The qRT-PCR validation of 10 DREB-A1 subgroup members under cold stress and hormone treatments showed that all *PgAP2/ERF* genes responded to cold stress but with distinct temporal patterns (Appendix A). The *PgAP2/ERF149* and *PgAP2/ERF272* peaked at 6 h post-treatment before declining significantly at 12 h, while *PgAP2/ERF185* and *PgAP2/ERF273* reached maximum expression at 12 h before decreasing at 24 h. In contrast, *PgAP2/ERF148*, *PgAP2/ERF150*, *PgAP2/ERF186*, *PgAP2/ERF187*, *PgAP2/ERF240* and *PgAP2/ERF241* showed continuously increasing expression levels, peaking at 24 h (Appendix A). Under hormone treatments, *PgAP2/ERF148*, *PgAP2/ERF149*, *PgAP2/ERF185*, *PgAP2/ERF241* and *PgAP2/ERF272* exhibited significantly elevated expression with GA treatment, while *PgAP2/ERF187* showed specific response to ABA treatment (Appendix A).

### 2.7. Functional Characterization of PgAP2/ERF187 in A. thaliana

To characterize the function of *PgAP2/ERF187*, the constructed overexpression vector pCAMBIA1300-PgAP2/ERF187 was transformed into Arabidopsis thaliana, yielding positive transgenic lines *PgAP2/ERF187*-OE2 and *PgAP2/ERF187*-OE5 (Figure 5). Cold tolerance assessment at the germination stage showed that under normal conditions, the germination rates of OE lines (OE-2 and OE-5) were comparable to Col-0, whereas exposure to 4 °C significantly reduced the germination rate across all lines (Figure 5a,c). Cold resistance tests indicated similar growth vigor between Col-0 and OE seedlings under normal conditions; however, during cold treatment, OE transgenic plants exhibited better growth performance and less wilting (Figure 5b,c). Physiological analysis revealed no significant differences in malondialdehyde (MDA) content, superoxide dismutase (SOD) activity, peroxidase (POD) activity, proline (Pro) content, or hydrogen peroxide (H_2_O_2_) levels between Col-0 and OE lines under normal conditions. Under cold stress, however, OE lines accumulated significantly less MDA and H_2_O_2_ and demonstrated markedly higher SOD and POD activities as well as Pro content compared to Col-0 (Figure 5d). Furthermore, the expression levels of cold-responsive genes *COR6.6*, *COR15A*, *COR47*, *COR78*, and *COR413IM1* were significantly upregulated in the overexpression lines under cold stress (Figure 5e).

### 2.8. Prediction and Functional Verification of PgAP2/ERF Downstream Target Genes

To investigate the molecular regulatory mechanism of *PgAP2/ERF* in cold stress response, we extracted 2000 bp promoter regions upstream of DREB-A1 subgroup member CDSs and mapped them to the *A. thaliana* transcription factor database to construct potential regulatory networks. The analysis revealed binding sites for ABF1 and ABF2 in the upstream region of *PgAP2/ERF187* (Figure 6a), consistent with promoter prediction results showing ABRE cis-elements (binding sites for ABF transcription factors) within the 2000 bp upstream region of *PgAP2/ERF187* (Appendix A). Through Blast alignment, we identified *EVM0000819* as the homologous gene of *ABF1/ABF2* in the *P. ginseng* genome and designated it as PgABF (Figure 6b). Subcellular localization demonstrated co-localization of PgAP2/ERF187 and PgABF in the nucleus (Figure 6c). Yeast one-hybrid assays confirmed that PgABF binds to the 2000 bp region upstream of *PgAP2/ERF187* CDS (Figure 6d). The results of dual-luciferase imaging and enzyme activity assays further support this conclusion (Figure 6e). These results indicate that PgABF likely regulates *PgAP2/ERF187* transcription by binding to ABRE elements in its promoter region.

## 3. Discussion

The AP2/ERF proteins represent a crucial class of plant transcription factors (TFs) known to extensively participate in various aspects of plant growth and stress responses, including drought stress, cold stress, salt stress, organ development, hormone biosynthesis, and disease resistance [28,29,30,31]. The *P. ginseng*, as a precious medicinal plant, possesses both significant pharmacological value and economic importance. Its bioactive components (e.g., ginsenosides) are widely applied in immunomodulation, anti-fatigue, and neuroprotection, with continuously growing global market demand. Although the functions of the AP2/ERF transcription factor family in stress responses and developmental regulation have been well characterized in model species such as *A. thaliana* [32] and *O. sativa* [33], systematic identification, structural characterization, and mechanistic exploration of the *P. ginseng* AP2/ERF gene family in stress resistance remain to be thoroughly investigated.

This study conducted a comprehensive genome-wide analysis of *P. ginseng*, identifying 318 *PgAP2/ERF* genes (Appendix A). Phylogenetic analysis using 494 AP2/ERF protein sequences (318 from *P. ginseng* and 176 from *A. thaliana*) revealed evolutionary characteristics and functional divergence of the *PgAP2/ERF* family. The family was classified into five subfamilies (Soloist, RAV, AP2, ERF, and DREB), with ERF and DREB grouped together due to their highly similar domains and further divided into 12 subgroups (A-1 to A-6 and B-1 to B-6) (Figure 1). This classification pattern aligns with studies in model plants like *A. thaliana* and *O. sativa* [34], demonstrating the evolutionary conservation of the AP2/ERF family in plants. Within the DREB-A1 subfamily phylogenetic analysis, we identified a group of highly conserved C-repeat/dehydration-responsive element binding factors (CBFs), including *A. thaliana CBF1* (*AT4G25490*), *CBF2* (*AT4G25470*), *CBF3* (*AT4G25480*), and *CBF4* (*AT5G51990*) (Figure 1). Numerous studies have confirmed that CBFs enhance plant cold adaptation by specifically binding to CRT/DRE cis-elements (core sequence: CCGAC) in target gene promoters, thereby activating downstream cold-responsive genes (e.g., *COR15A*, *RD29A*) [35,36]. For instance, in *A. thaliana*, *CBF1-3* regulates cold acclimation through the *ICE1*-dependent pathway, while *CBF4* participates in ABA-mediated cold response [37]. Therefore, *PgAP2/ERF* members within the same DREB-A1 subgroup likely share similar functions in cold stress resistance.

Domain differentiation serves as the key molecular basis for functional specialization. While all members contain the conserved AP2 domain, significant variations exist among subfamilies (Appendix A). The AP2 subfamily features dual AP2 domains and specific motifs (motif4/5/9) (Appendix A), with significantly more exons (6–10) than other subfamilies, potentially regulating developmental processes through transcriptional complexity, analogous to *A. thaliana* AP2 members’ functions [6]. The RAV subfamily contains B3 DNA-binding domains and may integrate ABA and auxin signaling in stress responses, like maize ZmbZIP4’s regulatory role in salt stress [38]. Subgroup-specific domains in DREB/ERF subfamilies (e.g., PHD SF, HepA, and Laccase domains in DREB-A2) may enhance stress adaptation through epigenetic modifications or redox regulation. For instance, PHD domains in PgAP2/ERF107/131/223 might resemble OsERF52’s phosphorylation modifications, activating cold-responsive genes via chromatin remodeling [33]. Notably, the Soloist subfamily’s four members (e.g., PgAP2/ERF85) contain unique PKc-like domains (Appendix A), potentially conferring regulatory functions independent of classical ERF/DREB pathways [39]. Gene structure complexity correlates with functional adaptation [40]. Most *PgAP2/ERF* members (85.22%) contain 1–2 exons, suggesting rapid transcriptional responses to stress, while AP2 subfamily’s multi-exon structures may enable precise developmental regulation through alternative splicing (Appendix A) [41]. Additionally, 30.5% of members possess UTR regions that may regulate translation efficiency via mRNA stability, akin to *CmABF1*’s post-transcriptional regulation in melon cold stress (Appendix A) [30].

Gene duplication analysis revealed that segmental duplication (400 pairs, accounting for 96.85%) is the main expansion mechanism of the *PgAP2/ERF* gene family, with its proportion significantly higher than that of tandem duplication (13 pairs) (Appendix A). This pattern aligns with *AP2/ERF* evolutionary trends in *A. thaliana* and *O. sativa* [42]. Notably, duplications clustered on specific chromosomes (e.g., Chr11/Chr14 with 19 duplicates each), while Chr15 showed minimal density (6) (Figure 2a), suggesting localized recombination or transposon activity [43]. Ka/Ks ratios (all <1) (Appendix A) indicate strong purifying selection, constraining functional divergence to preserve core stress-response functions [44], potentially linked to *AP2/ERF*’s pivotal roles in fundamental processes like *CBF*-mediated cold adaptation [37]. Interspecies synteny analysis revealed varying homology levels, with the highest conservation in *P. japonicus* (7 gene pairs) and the lowest in *P. stipuleanatus* (2) (Figure 2b), possibly reflecting post-speciation adaptive evolution, such as *P. stipuleanatus* plateau adaptation-driven *AP2/ERF* remodeling [45].

The AP2/ERF family plays a central role in plant stress responses, with members of the DREB subfamily participating in abiotic stress adaptation by regulating cold, drought, and salt stress-responsive genes [46]. For example, genes within the *A. thaliana* DREB-A1 subgroup—*CBF1/CBF2/CBF3*—serve as core regulators of cold stress response; their overexpression significantly activates the expression of COR (cold-regulated) genes and enhances cold tolerance [37,47,48]. In our study, 71.43% of DREB-subgroup PgAP2/ERF members showed significant responses to salt/cold stress, among which 10 DREB-A1 subgroup genes exhibited cold-specific induction (Figure 4a and Appendix A). These findings are highly consistent with the conserved role of DREB-A1 subgroup genes in cold response across species, further supporting the central importance of this subgroup in the evolutionary adaptation of plants to cold stress. Hormonal response characteristics exhibited divergence: 65.93% of DREB members responded to GA/ABA, including the ABA-specific regulator *PgAP2/ERF187* (Figure 4b and Appendix A). Similarly, *SlERF2* in tomato enhances cold tolerance by promoting ABA biosynthesis and activating CBF signaling [49]. The ABA-responsive nature of *PgAP2/ERF187* observed in our study aligns with these findings, suggesting a conserved mechanism whereby ERF genes mediate cold stress responses through ABA-dependent regulation. Furthermore, analogous to the discovery that a 60 bp InDel in *WRKY34* modulates chromatin plasticity to influence cold response in tomato [41], these results collectively highlight the conserved regulatory functions across different transcription factor families among species.

Functional validation using *PgAP2/ERF187*-overexpressing *A. thaliana* lines (OE-2 and OE-5) revealed its multi-layered cold adaptation mechanisms (Figure 5). Phenotypically, under 4 °C treatment, the OE lines exhibited higher germination rates and reduced wilting compared to Col-0 (Figure 5a–c), a response consistent with the cold-induced growth advantage and stability observed in transiently overexpressed *Tetrastigma hemsleyanum* ThERF46 lines [50], and like the cold tolerance phenotypes regulated by the maize CBL–CIPK pathway [51]. Physiologically, under cold stress, the OE lines showed decreased MDA content and increased SOD/POD activities (Figure 5d). Specifically, the OE lines accumulated significantly less MDA and H_2_O_2_ than Col-0, while demonstrating markedly higher SOD and POD activities as well as proline (Pro) content (Figure 5d). SOD acts as the first line of defense against ROS by dismutating superoxide anions (O_2_^−^∙) induced by cold stress into hydrogen peroxide (H_2_O_2_), while POD further decomposes H_2_O_2_ into harmless H_2_O and O_2_. Together, they maintain ROS homeostasis [52]. Proline enhances stress tolerance by regulating cellular osmotic pressure [53]. These coordinated changes in physiological indicators further demonstrate the role of *PgAP2/ERF187* in alleviating cold stress. Moreover, under cold stress, the expression levels of cold-responsive genes—*COR6.6*, *COR15A*, *COR47*, *COR78*, and *COR413IM1*—were significantly upregulated in the OE lines (Figure 5e). This aligns with the established mechanism in which *A. thaliana* CBF proteins enhance freezing tolerance by activating COR genes [54,55], suggesting that *PgAP2/ERF187* may participate in cold adaptation through regulating the expression of COR family genes. Heterologous expression of *MbCBF2* increased proline content and SOD/POD activities, reduced electrolyte leakage, and enhanced cold tolerance in *Malus baccata* [56], and *VvERF63* in grape reduces ROS accumulation by regulating antioxidant genes [57]. The physiological and molecular changes observed in our *PgAP2/ERF187*-OE lines are highly consistent with these functional characteristics of ERF genes, further supporting the central role of ERF family members in regulating ROS and activating downstream stress-responsive genes under cold stress.

In-depth analysis of the molecular interactions underlying ROS homeostasis regulation reveals that the ROS-modulating function of *PgAP2/ERF187* is closely coupled with ABA signaling. Previous studies have shown that ABA can activate ABF transcription factors, which in turn regulate the expression of both ERF genes and ROS-scavenging genes, forming a coordinated “ABA–ABF–ROS scavenging” pathway [58]. In this study, ABRE *cis*-elements in the promoter region of *PgAP2/ERF187* matched predicted ABF binding sites (Figure 6a). Homology analysis confirmed that PgABF (EVM0000819) is an ortholog of *A. thaliana* ABF1/2 (Figure 6b) [59,60]. Nuclear co-localization (Figure 6c), yeast one-hybrid assay (Figure 6d), and dual-luciferase complementation assay (Figure 6e) further verified that PgABF can directly bind to this promoter (Figure 6d), establishing an “ABA–ABF–PgAP2/ERF187” regulatory hierarchy. Combined with physiological data, we speculate that PgABF may function through dual regulatory mechanisms: on the one hand, it directly activates the expression of *PgAP2/ERF187*; on the other hand, it may synergize with *PgAP2/ERF187* to co-regulate downstream ROS-scavenging enzyme genes (such as SOD and POD) and *COR* genes—a mechanism similar to that observed in rice, where OsABF1 forms a complex with OsERF71 to bind the promoter of a POD gene and enhance its expression [61]. Moreover, the canonical ABF–ICE1–CBF signaling cascade in cold adaptation also relies on the direct regulation of downstream transcription factors by ABFs [62,63,64]. The “ABA–ABF–PgAP2/ERF187” pathway identified in our study echoes these established mechanisms, not only refining the ABA-mediated transcriptional regulatory network in cold response but also providing new evidence supporting the existence of an ABA–ERF–ROS regulatory axis and an ABA–ERF–COR regulatory axis.

## 4. Materials and Methods

### 4.1. Identification and Analysis of PgAP2/ERF Gene Family in P. ginseng

The genome and protein data of *P. ginseng* were downloaded from the National Genomics Data Center (https://ngdc.cncb.ac.cn/, accessed on 12 June 2025) [45], while the hidden Markov model (PF00847) for AP2 genes was obtained from the Pfam database (http://pfam.xfam.org/,accessed on 12 June 2025). The *P. ginseng* genome was searched using HMMER 3.2.1 software (E-value cutoff: 1 × 10^−5^), and sequences with complete LOB domains were further screened through the NCBI Conserved Domain Database (https://www.ncbi.nlm.nih.gov/cdd/, accessed on 12 June 2025) and SMART database (http://smart.embl-heidelberg.de/, accessed on 12 June 2025) to identify PgAP2/ERF family members for subsequent analysis. Subcellular localization was predicted using WoLF PSORT (https://www.genscript.com/wolf-psort.html, accessed on 12 June 2025), while physicochemical parameters were predicted via ExPASy ProtParam tool (https://web.expasy.org/protparam/, accessed on 12 June 2025).

### 4.2. Phylogenetic Analysis of PgAP2/ERF Family Genes

The *A. thaliana* AP2/ERF protein sequences were retrieved from TAIR (https://www.arabidopsis.org/, accessed on 12 June 2025) and aligned with PgAP2/ERF members using MAFFT (http://mafft.cbrc.jp/alignment/software/, accessed on 12 June 2025) under default parameters. The maximum-likelihood phylogenetic tree was constructed with IQ-TREE based on the JTTDCMut + F + R4 model [65], with branch support evaluated through 1000 bootstrap replicates. Visualization and annotation were performed using iTOL (https://itol.embl.de/, accessed on 12 June 2025).

### 4.3. Analysis of Gene Structure, Conserved Domains, and Cis-Acting Elements in PgAP2/ERF Family

Conserved motifs in PgAP2/ERF proteins were identified via MEME v5.5.5 suite (https://meme-suite.org/meme/doc/meme.html, accessed on 12 June 2025) [66]. Gene structure information (exon/intron coordinates) was analyzed using GFF annotation files. The 2000 bp upstream sequences of transcription start sites (TSS) were extracted to predict cis-regulatory elements using PlantCARE (http://bioinformatics.psb.ugent.be/webtools/plantcare/html/, accessed on 12 June 2025) [67].

### 4.4. Duplication and Synteny Analysis of PgAP2/ERF Genes

The *A. thaliana* genome data were obtained from Phytozome v13 (https://phytozome-next.jgi.doe.gov/, accessed on 12 June 2025), while *P. quinquefolius*, *P. stipuleanatus*, and *P. japonicus* genomes were sourced from NGDC [45]. MCScanX (E-value: 1 × 10^−5^) was employed for intra- and inter-species synteny analysis to identify orthologous/paralogous relationships. Ka/Ks ratios were calculated using KaKs_Calculator 2.0 to assess selection pressures.

### 4.5. RNA-Seq and Expression Analysis

RNA-seq data under various stress treatments were acquired from the *P. ginseng* Genome Database (http://ginsengdb.snu.ac.kr/, accessed on 12 June 2025) [68], while hormone treatment data came from Wang et al. [69]. Data were preprocessed and normalized to TPM values, with heatmaps generated using R v4.4.2.

### 4.6. RNA Extraction and qRT-PCR Analysis

Total RNA was extracted from 2-week-old *P. Ginseng* seedlings treated at 4 °C for 6/12/24 h or with different hormones (ABA: 50 mM, IAA: 10 mM, 6-BA: 75 mM, GA_3_: 100 mM, with distilled water treatment as the control, for 5 h; specific methods refer to Wang et al. [70]) using the RNAprep pure Plant Kit (TIANGEN, Beijing, China), followed by cDNA synthesis. Then, 3 μL of total RNA was used for genomic DNA removal and cDNA synthesis with the TransScript^®^ One-Step gDNA Removal and cDNA Synthesis SuperMix kit (TransGen Biotech, Beijing, China). Quantitative real-time PCR was performed with gene-specific primers (Appendix A) using PerfectStart^®^ qPCR SuperMix (TransGen Biotech, Beijing, China) on a LightCycler 96 system (ROCHE, Basle, Switzerland). The *β-actin* gene was used as the internal control, and relative expression levels were calculated using the 2^−ΔΔCT^ method [71].

### 4.7. Subcellular Localization

The *PgAP2/ERF187* CDS was cloned into the *PHB-YFP* vector (*HindIII*/*SacI*-digested; primers in Appendix A). Recombinant plasmids were transformed into Agrobacterium GV3101 for tobacco leaf infiltration. After infiltration, the plants were kept in the dark at 25 °C for 12 h, then transferred to normal light conditions for 2–3 days. YFP fluorescence signals were finally observed using confocal microscopy.

### 4.8. Plant Material and Treatments

The *A. thaliana* (Col-0) seeds (Shaanxi Aiyouji Biotech Co., Xi’an, China) were surface-sterilized (75% ethanol), vernalized (4 °C, dark, 3 d), and grown in soil under controlled conditions (26 °C/20 °C, 16/8 h light/dark, 75% RH) until flowering.

### 4.9. Overexpression of PgAP2/ERF187 in A. thaliana

The *PgAP2/ERF187* gene was cloned into the *pCAMBIA1300* vector (primers listed in Appendix A) and introduced into GV3101. The floral dip method was employed for A. thaliana transformation. Transgenic plants were selected on 1/2 MS medium containing 25 mg/L hygromycin. Homozygous T3 lines were identified through PCR-based genotyping and hygromycin resistance segregation analysis in the T2 and T3 generations. Two overexpression lines, *PgAP2/ERF187*-OE2 and *PgAP2/ERF187*-OE5, were selected for further study based on their high and stable transgene expression levels as determined by qRT-PCR analysis (Appendix A). T3 transgenic and wild-type seeds were germinated on 1/2 MS medium for subsequent phenotypic assays.

### 4.10. Phenotypic Analysis of PgAP2/ERF187-Transformed A. thaliana

To evaluate the germination rate of T3-generation transgenic *A. thaliana* under cold stress, seeds of wild-type and two transgenic lines were sown in sterilized soil and incubated at 4 °C. Germination rates were recorded after 10 days of cultivation. Seedlings were then transferred to media maintained at 4 °C and 23 °C, respectively, for vertical cultivation over 14 days to measure primary root growth and observe phenotypes. Furthermore, Col and OE lines of *A. thaliana* were subjected to 72 h treatment in a 4 °C growth chamber for phenotypic observation.

### 4.11. Physiological/Biochemical Assays in Transgenic A. thaliana

Three-week-old control plants and two overexpression transgenic lines (OE-2 and OE-5) were subjected to a 24 h cold stress treatment at 4 °C. Subsequently, physiological and biochemical parameters were measured, including proline (Pro) content, malondialdehyde (MDA) content, superoxide dismutase (SOD) activity, hydrogen peroxide (H_2_O_2_) content, and peroxidase (POD) activity.

Pro content was determined using the sulfosalicylic acid extraction–ninhydrin colorimetric method; MDA content was measured according to the thiobarbituric acid (TBA) method; SOD activity was assessed by the nitroblue tetrazolium (NBT) photochemical reduction method; H_2_O_2_ content was quantified via the titanium sulfate colorimetric method; and POD activity was measured using the guaiacol method at a wavelength of 470 nm. All experimental procedures were performed following previously described methods [72,73], with three biological replicates per sample.

### 4.12. Transcriptional Regulatory Network Prediction

The 2000 bp upstream sequences of the CDS of the extracted *PgAP2/ERF* genes were submitted to the Regulation Prediction module of the Plant Transcriptional Regulatory Map database (http://plantregmap.gao-lab.org/, accessed on 12 June 2025) to predict potential transcription factors that may interact with the input sequences (E-value < 1 × 10^−5^) [74]. The analysis results were downloaded locally, and the transcription factor regulatory network was constructed using Cytoscape v3.10.3 software.

### 4.13. Yeast One-Hybrid Assay and Luciferase Complementation Assay

The *PgABF* CDS and *PgAP2/ERF* promoter (0–2000 bp) were cloned into *pGADT7* and *pHis2* vectors (primers in Appendix A), then co-transformed into Y187 yeast. Transformants were selected on SD/-Trp/-Leu and SD/-Trp/-Leu/-His + 30 mM 3-AT plates, with p53 system controls.

For the dual-luciferase reporter assay, the coding sequence (CDS) of PgABF was cloned into the *pGreenII 62-SK* vector, and the promoter fragment of PgAP2/ERF187, spanning 0–2000 bp upstream of the start codon, was cloned into the *pGreenII 0800-LUC* vector (primers are provided in Appendix A) [75]. The recombinant vectors were transformed into the GV3101 strain and injected into tobacco leaves. LUC fluorescence signals were detected using the GloMax 20/20 luminometer (Promega, Madison, WI, USA) chemiluminescence imaging system, and luciferase activity was measured with luciferin as the substrate.

### 4.14. Statistical Analysis

Gene expression data (under cold/hormone treatments) were analyzed using one-way ANOVA (*n* = 3, *p* ≤ 0.01), while phenotypic, biochemical parameters and cold-related gene expression statistics (*n* = 3, *** *p* ≤ 0.001) were subjected to t-test analysis using the latest version of GraphPad Prism V10.1.

## 5. Conclusions

This study identified 318 PgAP2/ERF family members in *P. ginseng* through genome-wide analysis and classified them into five subfamilies (AP2, DREB, ERF, RAV, and Soloist) based on phylogenetic relationships. Gene duplication analysis indicated that segmental duplication served as the primary driver for the expansion of this family, with most genes being conserved under purifying selection. Expression profiling revealed that all members of the DREB-A1 subgroup responded significantly to cold stress, with *PgAP2/ERF187* being specifically induced by ABA. Overexpression of *PgAP2/ERF187* in *A. thaliana* markedly enhanced cold tolerance, as evidenced by improved seed germination rates, reduced wilting, significantly decreased proline and MDA content, and increased H_2_O_2_ levels along with elevated SOD and POD activities under cold stress—indicating its role in enhancing ROS scavenging capacity to alleviate oxidative damage. Mechanistic investigations confirmed the nuclear co-localization of PgABF (the *P. ginseng* ortholog of *A. thaliana* ABF1/2) with PgAP2/ERF187 and demonstrated that PgABF directly binds to ABRE *cis*-elements in the *PgAP2/ERF187* promoter region to regulate its transcription. Collectively, this study elucidates for the first time the molecular mechanism by which ABA signaling mediates cold adaptation in *P. ginseng* through the “PgABF–PgAP2/ERF187” regulatory module, providing critical theoretical insights into cold resistance and valuable genetic resources for breeding cold-tolerant cultivars.

## Figures and Tables

**Figure 1 plants-14-02922-f001:**
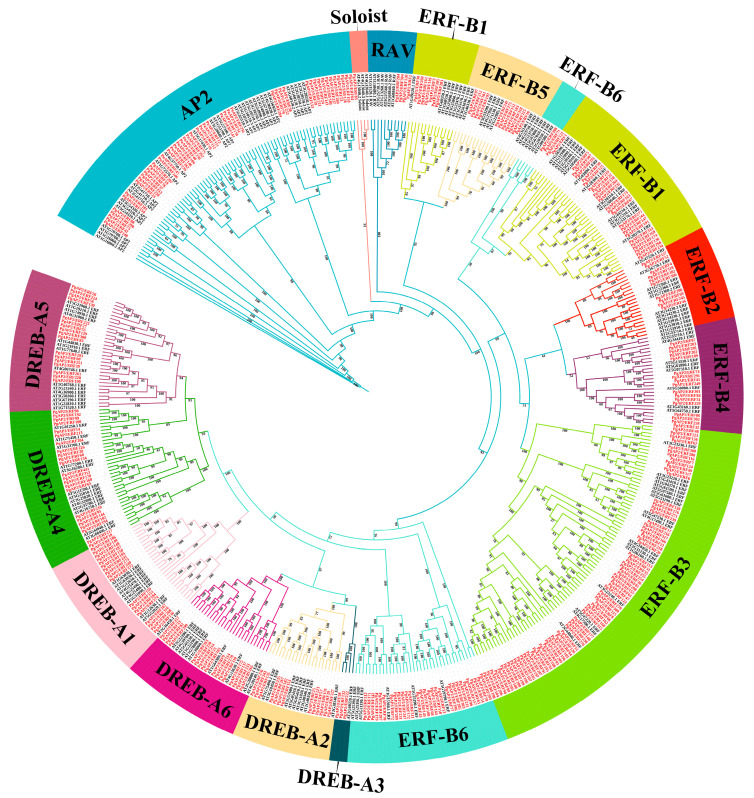
Phylogenetic tree of AP2/ERF proteins from *Panax ginseng* C. A. Meyer (*P. ginseng*) and *Arabidopsis thaliana* (*A. thaliana*). The phylogenetic tree was constructed using the maximum likelihood method in IQ-TREE based on the JTTDCMut + F + R4 model. The AP2/ERF proteins from *P. ginseng* are highlighted in red.

**Figure 2 plants-14-02922-f002:**
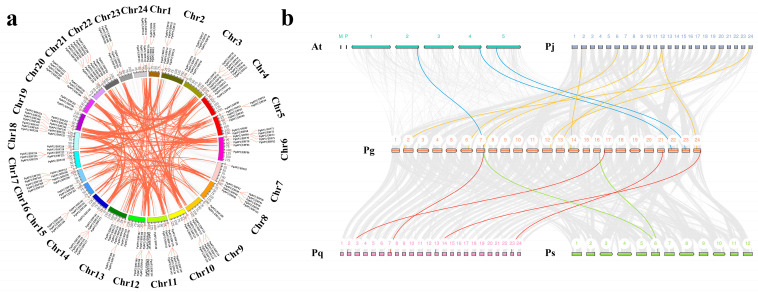
Chromosomal distribution and evolutionary relationship analysis of *PgAP2/ERF* gene family members in *P. ginseng*. (**a**) Intraspecies collinearity of *PgAP2/ERF* genes. Red lines indicate duplicated *PgAP2/ERF* gene pairs. (**b**) Interspecies collinearity between *Panax ginseng* (Pg) and *Arabidopsis thaliana* (At), *Panax japonicus* (Pj), *Panax quinquefolium* (Pq), and *Panax stipuleanatus* (Ps). Different colors represent duplicated *PgAP2/ERF* pairs between *P. ginseng* and different species.

**Figure 3 plants-14-02922-f003:**
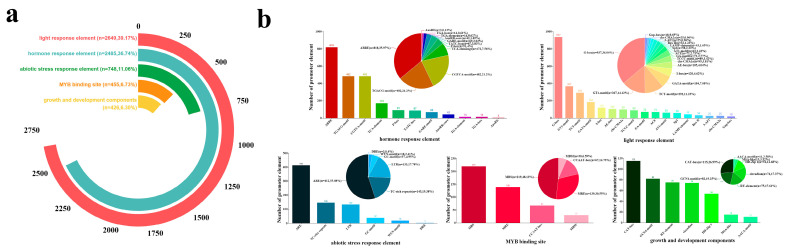
Composition and Distribution Characteristics of Cis-Acting Elements in the *PgAP2/ERF* Gene Family. (**a**) Proportional distribution of five major types of cis-acting elements. (**b**) Element-specific proportional distribution across different categories.

**Figure 4 plants-14-02922-f004:**
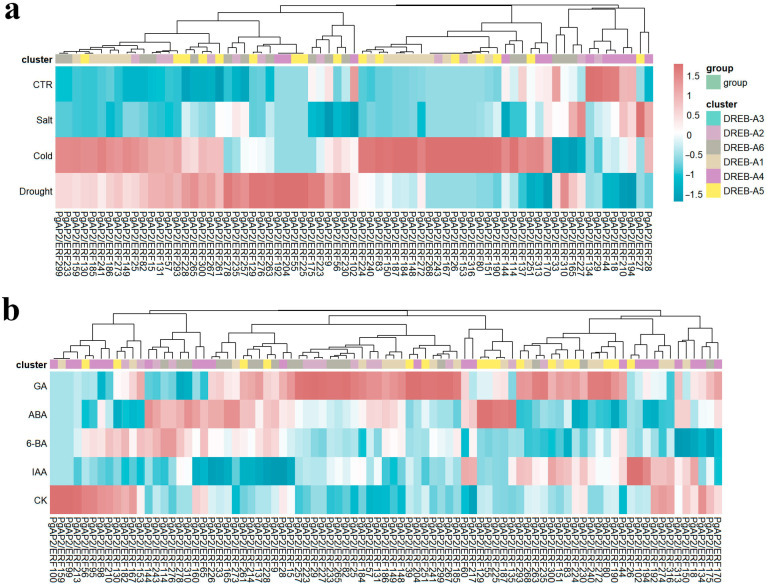
Expression patterns of *PgAP2/ERF* genes in the DREB subgroup. (**a**) Expression levels of *PgAP2/ERF* genes under different abiotic stresses. (**b**) Expression levels of *PgAP2/ERF* genes under different hormone treatments.

**Figure 5 plants-14-02922-f005:**
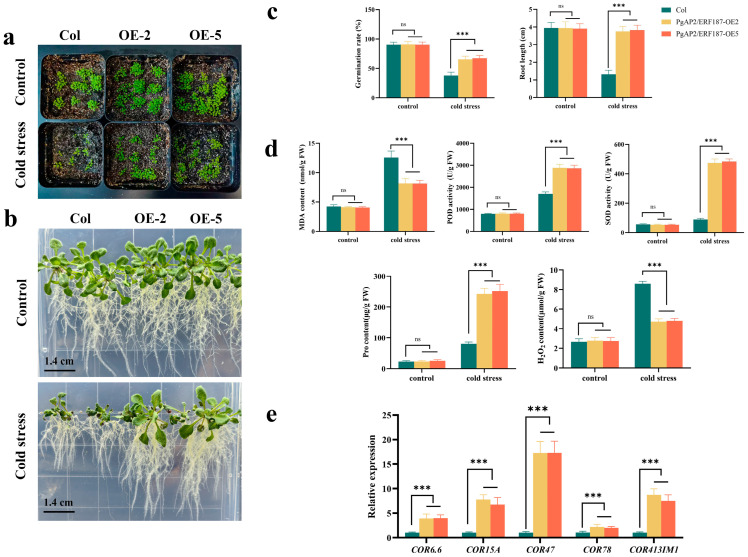
Functional analysis of *PgAP2/ERF187* transgenic *A. thaliana* under cold stress. (**a**) Germination phenotypes of wild-type (Col) and *PgAP2/ERF187* overexpression lines (*PgAP2/ERF187*-OE2 and *PgAP2/ERF187*-OE5; OE represents Overexpression lines). (**b**) Phenotypes of 2-week-old Col and transgenic plants. Scale bar: 1.4 cm. (**c**) Statistical analysis of germination rates and primary root lengths in Col and transgenic plants (cold stress treatment: 4 °C for 72 h). (**d**) Comparison of physiological and biochemical parameters between Col and transgenic *A. thaliana* after cold stress treatment (cold stress treatment: 4 °C for 24 h). (**e**) Expression levels of cold-resistant genes in Col and transgenic plants (cold stress treatment: 4 °C for 24 h). Note: Values represent mean ± SE (*n* = 3); asterisks denote significant differences as determined by *t*-test analysis (*** *p* ≤ 0.001), “ns” indicates not significant (*p* > 0.05).

**Figure 6 plants-14-02922-f006:**
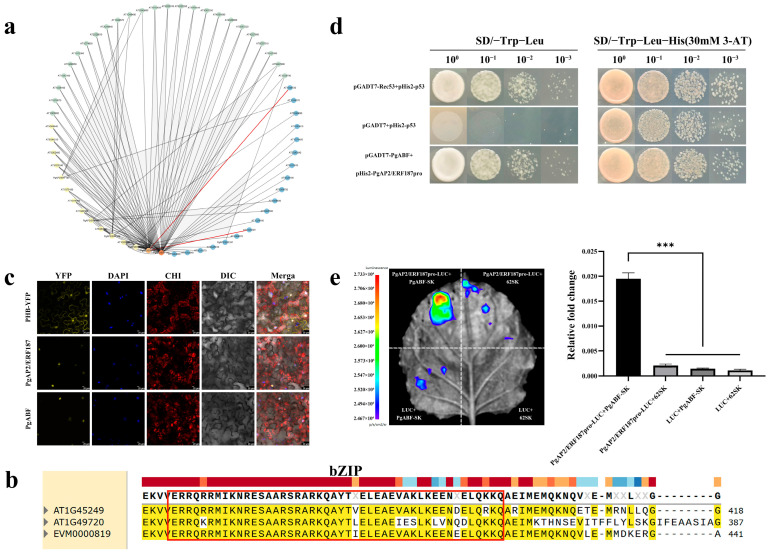
PgABF Binding to the *PgAP2/ERF187* Promoter. (**a**) Regulatory network analysis of DREB-A1 subgroup members in *PgAP2/ERF* family. (**b**) Homology alignment of ABF1/ABF2 orthologs in the *P. ginseng* genome. (**c**) Subcellular localization of PgABF and PgAP2/ERF187. (**d**) Y1H validation. (**e**) Dual-luciferase imaging and enzyme activity detection. Note: The region within the red box represents the conserved bZIP domain sequence. The values represent mean ± SE (*n* = 4); asterisks denote significant differences as determined by *t*-test analysis (*** *p* ≤ 0.001).

## Data Availability

Data are contained within the article and Appendix A.

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
