# Peer review of "Genome-Wide Identification of the AP2/ERF Gene Family and Functional Analysis of PgAP2/ERF187 Under Cold Stress in Panax ginseng C. A. Meyer"

_plants, 2025, doi:10.3390/plants14182922_

Round 1

Reviewer 1 Report

Comments and Suggestions for Authors

The manuscript is interesting and it will be fundamental for future functional analysis and cold stress tolerance improvement. However, there are several issues that must be addressed before publication.

  • It is known that ERF contain the AP2 domain. The name of each gene can be revised as PgERF1-PgERF318, AP2 may not be needed for the names.
  • Since the manuscript characterized the function of ERF187 based on physiological and biochemical traits, but these traits are nor mentioned in the introduction. Here authors are suggested to incorporate how cold stress regulate these traits and how plants cope with cold stress; a focus on physiological and biochemical alterations.
  • The font size of all figures must be increased; it is hard to see the names. For figure 4a, b increase the width so that we can clearly see the name of the genes.
  • For qRT-qPCR analysis use letters for each bar to differentiate the significance differences for clarity. Also mention the data analysis test used. ABA treatment has been mentioned twice in the graph; correct it with the right hormone.
  • The detailed methods for hormone treatments, the concentration of hormones must be described in the method.
  • It is interesting to see that ERF187 overexpression enhanced Arabidopsis cold tolerance, while the results are quite limited to conclude. Here authors are suggested to measure the expression of some key cold-related marker genes in Arabidopsis, so that we can at least partially see the molecular mechanisms how ERF187 works in cold stress improvement.
  • For the regulation of PgAP2/ERF187 by PgABF, I suggest the authors to perform transactivation assay in tobacco cells, to see if ABF can directly activate the promoter of ERF187, otherwise it is hard to conclude this result only from yeast one hybrid assay.
  • All methods are not sufficiently described. Add the details of cDNA synthesis process and kit, the amount of RNA used. Add the reference for the qPCR analysis method.
  • Provide incubation period, light and dark conditions after infiltration for subcellular localization.
  • Add the detailed procedure for Arabidopsis transformation and homozygous lines identification, including the method used. Describe how the two overexpression lines were selected for further study, if you have expression data for the two lines and others provide it.
  • For seed germination and root analysis authors used 72h of cold stress treatment, but adult plants used for biochemical assays, only 24h is used. Is 24h sufficient to induce cold stress for adult plants? please explain in the text why you use two different times for cold treatment and even a shorter time for adult plants.
  • Despite a reference is provided for the methods of Physiological/Biochemical Assays, still authors are required to add some detailed procedures for each assays.
  • Regarding data analysis, not all results presented in this manuscript used one type of test to see the significance levels. Authors are suggested to add more information on statistical analysis, including the tests used for each result section.
  • The discussion lacks some key references, especially gene functional studies on cold stress response and associated mechanisms. Authors are suggested to add some previous studies particularly gene functions and relate with the present study.
Comments on the Quality of English Language
  • English language should be revised by native speaker, there are several grammar errors.

Author Response

Dear Reviewer:

We would like to submit the enclosed manuscript entitled “Genome-Wide Identification of the AP2/ERF Gene Family and Functional Analysis of PgAP2/ERF187 under Cold Stress in Panax ginseng” (ID: plants-3837814). We sincerely appreciate the reviewer’s unwavering patience and rigorous guidance throughout the review process of our manuscript. Your dedication to ensuring the quality of every submission is truly admirable, and we are deeply grateful for the time and expertise you have devoted to evaluating our research.

We are deeply grateful to the reviewers for their invaluable suggestions, which have provided critical insights to further strengthen our manuscript. The reviewers’ expertise has guided us to refine both the methodological rigor and the clarity of our findings. In response to their comments, we have carefully revised the manuscript and sought the assistance of a professional native English-speaking reviewer to thoroughly polish the language, ensuring clarity, fluency, and adherence to academic conventions. We have humbly adopted each suggestion, and all revisions have been systematically documented in the text below, where we address every comment with supporting evidence and rationale.

We sincerely hope the revised manuscript now meets the exceptional standards of Plants-Basel. Should any additional clarifications be needed, we pledge to respond promptly and thoroughly.

Reviewer 1

  1. It is known that ERF contains the AP2 domain. The name of each gene can be revised as PgERF1-PgERF318, AP2 may not be needed for the names.

We sincerely thank the reviewer for this valuable suggestion, and we highly respect their opinion. Regarding the gene naming convention, we referred to the classification criteria provided in PlantTFDB v5.0 (https://planttfdb.gao-lab.org/). The AP2/ERF superfamily is defined by the AP2/ERF domain, which consists of approximately 60–70 amino acids and is involved in DNA binding. It is further divided into three families: AP2 family proteins contain two repeated AP2/ERF domains; ERF family proteins contain a single AP2/ERF domain; and RAV family proteins contain a B3 domain—a DNA-binding domain conserved in other plant-specific transcription factors—in addition to a single AP2/ERF domain. Since our study addresses the classification and identification of the entire AP2/ERF superfamily rather than focusing solely on the ERF subfamily, we retained "AP2" in the nomenclature to accurately reflect the systematic classification attributes. This approach is also consistent with widely accepted naming conventions in the field. We have also cited relevant literature that supports this naming method.

[1] Zhang H, Wang S, Zhao X, Dong S, Chen J, Sun Y, Sun Q, Liu Q: Genome-wide identification and comprehensive analysis of the AP2/ERF gene family in Prunus sibirica under low-temperature stress. BMC plant biology 2024, 24(1):883.

[2] Zhang M, Lu P, Zheng Y, Huang X, Liu J, Yan H, Quan H, Tan R, Ren F, Jiang H et al: Genome-wide identification of AP2/ERF gene family in Coptis Chinensis Franch reveals its role in tissue-specific accumulation of benzylisoquinoline alkaloids. BMC genomics 2024, 25(1):972.

[3] Wang H, Ni D, Shen J, Deng S, Xuan H, Wang C, Xu J, Zhou L, Guo N, Zhao J et al: Genome-Wide Identification of the AP2/ERF Gene Family and Functional Analysis of GmAP2/ERF144 for Drought Tolerance in Soybean. Frontiers in plant science 2022, 13:848766.

[4] Faraji S, Filiz E, Kazemitabar SK, Vannozzi A, Palumbo F, Barcaccia G, Heidari P: The AP2/ERF Gene Family in Triticum durum: Genome-Wide Identification and Expression Analysis under Drought and Salinity Stresses. Genes 2020, 11(12).

[5] Wan R, Song J, Lv Z, Qi X, Han X, Guo Q, Wang S, Shi J, Jian Z, Hu Q et al: Genome-Wide Identification and Comprehensive Analysis of the AP2/ERF Gene Family in Pomegranate Fruit Development and Postharvest Preservation. Genes 2022, 13(5).

  1. Since the manuscript characterized the function of ERF187 based on physiological and biochemical traits, but these traits are nor mentioned in the introduction. Here authors are suggested to incorporate how cold stress regulate these traits and how plants cope with cold stress; a focus on physiological and biochemical alterations.

We sincerely thank the reviewer for this insightful suggestion. As recommended, we have now supplemented the Introduction (Lines 56–63) with a description of the physiological and biochemical alterations induced by cold stress and how plants respond to these changes. The added content emphasizes cold-induced membrane rigidification, reactive oxygen species (ROS) accumulation, and the role of antioxidant systems and osmoprotectant accumulation in mitigating cold damage. Additionally, we have further strengthened the correlation between CBF-regulated cold tolerance and its physiological mechanisms in Line 73-75 by highlighting that CBF enhances cold tolerance through elevating antioxidant enzyme activity and osmolyte accumulation. These modifications are supported by corresponding references.

  1. The font size of all figures must be increased; it is hard to see the names. For figure 4a, b increase the width so that we can clearly see the name of the genes.

We sincerely appreciate the reviewer’s valuable feedback. In response to the comment, we have increased the font sizes in all figures to improve readability, except for Figure 1. Due to the large number of gene members included in the phylogenetic tree, further enlargement of gene labels would result in overlapping text and reduce clarity. Additionally, as recommended, we have specifically expanded the width of Figure 4a and 4b to ensure that gene names are clearly visible. These adjustments were implemented with the support of MDPI Author Services during the typesetting process to meet formatting requirements.

  1. For qRT-qPCR analysis use letters for each bar to differentiate the significance differences for clarity. Also mention the data analysis test used. ABA treatment has been mentioned twice in the graph; correct it with the right hormone.

We sincerely thank the reviewer for these insightful suggestions. We have carefully revised the manuscript accordingly. Specifically, in the qRT-qPCR analysis, the significance indicators in the bar charts have been changed from asterisks to letters to enhance clarity, and the statistical method has been updated from the t-test to one-way ANOVA with appropriate post hoc tests. Additionally, the hormone name that was incorrectly labeled as "ABA" in the figure has been corrected to "IAA". These modifications improve the accuracy and readability of our results.

  1. The detailed methods for hormone treatments, the concentration of hormones must be described in the method.

We thank the reviewer for this helpful suggestion. As recommended, we have now added detailed hormone treatment methods and specific concentration information to Section 4.6 of the Materials and Methods. The revised text clearly states hormone treatments were applied using ABA: 50 mM, IAA: 10 mM, 6-BA: 75 mM, and GA₃: 100 mM, with distilled water as the control, for a duration of 5 hours in two-week-old seedlings.

  1. It is interesting to see that ERF187 overexpression enhanced Arabidopsis cold tolerance, while the results are quite limited to conclude. Here authors are suggested to measure the expression of some key cold-related marker genes in Arabidopsis, so that we can at least partially see the molecular mechanisms how ERF187 works in cold stress improvement.

We sincerely thank the reviewer for this insightful suggestion. As recommended, we measured the expression of key cold-responsive marker genes in Arabidopsisto further explore the potential molecular mechanisms by which ERF187 enhances cold tolerance. Based on literature, the ICE1-CBF-COR pathway is recognized as the most important and typical cold stress response pathway in plants; however, its transcriptional activation involves a complex network of regulation, including transcription factors, post-translational modifications, light signaling, circadian factors, and interacting proteins [6]. Accordingly, using RT-qPCR, we observed elevated expression levels of cold-regulated (COR) genes—including COR6.6, COR413IM1, COR78, COR47, and COR15A—in ERF187-overexpressing lines (Figure 6), providing further insight into how ERF187 may contribute to cold stress adaptation.

[6] Qian Z, He L, Li F: Understanding cold stress response mechanisms in plants: an overview. 2024, Volume 15 - 2024.

  1. For the regulation of PgAP2/ERF187 by PgABF, I suggest the authors to perform transactivation assay in tobacco cells, to see if ABF can directly activate the promoter of ERF187, otherwise it is hard to conclude this result only from yeast one hybrid assay.

We sincerely thank the reviewer for this valuable suggestion. As recommended, we have performed the dual-luciferase complementation assay in tobacco cells to further validate the regulation of PgAP2/ERF187by PgABF. The results (Figure 7) demonstrate that PgABF can directly activate the promoter of PgAP2/ERF187, providing stronger evidence for the regulatory relationship between them. These additions have been incorporated into subsection 2.8 of the Results, section 4.13 of the Methods, and the legend of Figure 7. We believe this additional experiment significantly strengthens our conclusion.

  1. All methods are not sufficiently described. Add the details of cDNA synthesis process and kit, the amount of RNA used. Add the reference for the qPCR analysis method.

We sincerely thank the reviewer for this valuable suggestion. As recommended, we have now supplemented the details of the cDNA synthesis process in Section 4.6 of the Methods, including the specification of the TransScript® One-Step gDNA Removal and cDNA Synthesis SuperMix kit and the exact amount of RNA (3 μL) used for the reaction. Additionally, the reference for the qPCR analysis method has been added [7]. We believe these modifications have significantly improved the clarity and reproducibility of our methodological description.

[7] Livak KJ, Schmittgen TD: Analysis of relative gene expression data using real-time quantitative PCR and the 2(-Delta Delta C(T)) Method. Methods (San Diego, Calif) 2001, 25(4):402-408.

  1. Provide incubation period, light and dark conditions after infiltration for subcellular localization.

We sincerely thank the reviewer for pointing out this important omission. As suggested, we have now included detailed incubation conditions after infiltration in the revised manuscript (Lines 483–485). The added sentence states: “After infiltration, the plants were kept in the dark at 25 °C for 12 hours, then transferred to normal light conditions for 2–3 days. YFP fluorescence signals were finally observed using confocal microscopy.” We believe this clarification helps improve the reproducibility and completeness of the method description.

  1. Add the detailed procedure for Arabidopsis transformation and homozygous lines identification, including the method used. Describe how the two overexpression lines were selected for further study, if you have expression data for the two lines and others provide it.

We sincerely thank the reviewer for this suggestion. We have now added detailed procedures for Arabidopsis transformation and homozygous line identification in the Methods section. Specifically: Arabidopsis transformation was performed using the floral dip method with GV3101; Transgenic plants were selected on ½ MS medium containing 25 mg/L hygromycin; Homozygous T3 lines were identified through PCR-based genotyping and hygromycin resistance segregation analysis across T2 and T3 generations. The two overexpression lines, PgAP2/ERF187‐OE2and PgAP2/ERF187‐OE5, were selected based on their high and stable transgene expression levels, as determined by qRT‐PCR analysis (see Supplementary Figure 2).

  1. For seed germination and root analysis authors used 72h of cold stress treatment, but adult plants used for biochemical assays, only 24h is used. Is 24h sufficient to induce cold stress for adult plants? please explain in the text why you use two different times for cold treatment and even a shorter time for adult plants.

Thank you for your valuable question. The differential setting of cold treatment durations is based on the physiological characteristics of plants at different growth stages and the response patterns of the detection indicators. Seed germination and root development are fundamental processes in the early growth of plants, and their responses to cold stress exhibit a cumulative effect. Studies have shown that seed germination involves a series of physiological events such as endosperm decomposition and radicle breakthrough of the seed coat, and the inhibitory effect of cold stress on these processes takes a certain time to stably manifest (usually 48-72 hours) [8]. We chose 72 hours of treatment to more accurately reflect the long-term effects of cold stress on morphological indicators such as seed germination rate and root length, which is consistent with the treatment duration in most seed physiological studies [9]. For the biochemical analysis of adult plants, we considered that biochemical reactions in plants under cold stress, such as oxidative stress and accumulation of osmotic regulatory substances, have the characteristic of rapid response. Existing studies have confirmed that within 24 hours after cold stress, the ROS level, activities of antioxidant enzymes (such as SOD and POD), and proline content in plant cells have undergone significant changes and reached a detectable stable threshold [10, 11]. If the treatment time is extended, adult plants may initiate adaptive regulatory mechanisms (such as up-regulation of gene expression), which would instead mask the biochemical characteristics in the initial stress state. Therefore, 24-hour treatment is more suitable for capturing key early biochemical responses to cold stress.

[8] Bewley J D. Seed germination and dormancy [J]. Plant Cell, 1997, 9 (7): 1055-1066.

[9] Li Y, et al. Cold stratification duration affects seed germination characteristics of 12 alpine herbs from the eastern Qinghai-Tibet Plateau [J]. Ecology and Evolution, 2021, 11 (13): 8802-8813.

[10] Zhang Y, et al. Time-course analysis of physiological and transcriptomic responses to cold stress in Medicago truncatula [J]. BMC Plant Biology, 2022, 22 (1): 349.

[11] Wang L, et al. Short-term cold stress induces differential physiological and transcriptomic responses in two contrasting maize inbred lines [J]. Frontiers in Plant Science, 2020, 11: 587634.

  1. Despite a reference is provided for the methods of Physiological/Biochemical Assays, still authors are required to add some detailed procedures for each assays.

We sincerely thank the reviewer for this valuable suggestion. We have supplemented detailed methods for each physiological and biochemical assay in Section 4.11 of the Materials and Methods section and provided corresponding references to ensure the reproducibility of the methods. “Pro content was determined using the sulfosalicylic acid extraction–ninhydrin colorimetric method; MDA content was measured according to the thiobarbituric acid (TBA) method; SOD activity was assessed by the nitroblue tetrazolium (NBT) photochemical reduction method; H₂O₂ content was quantified via the titanium sulfate colorimetric method; and POD activity was measured using the guaiacol method at a wavelength of 470 nm. All experimental procedures were performed following previously described methods, with three biological replicates per sample.”

  1. Regarding data analysis, not all results presented in this manuscript used one type of test to see the significance levels. Authors suggest adding more information on statistical analysis, including the tests used for each result section.

We fully agree with the reviewer on the importance of clarifying statistical tests. As suggested, we have now comprehensively detailed the statistical methods applied to each figure and result in the “Statistical Analysis” section (4.14). For multi-group comparisons (e.g., under different treatments or time points), one-way ANOVA was used to determine whether significant differences exist between at least two groups, followed by appropriate post-hoc tests. For direct pairwise comparisons between a treatment group and the control, independent samples t-test was applied. All analysis was performed using the latest version of GraphPad Prism, and exact p-values or significance levels are reported in respective figure legends. These modifications ensure improved transparency, reproducibility, and statistical accuracy throughout the manuscript.

  1. The discussion lacks some key references, especially gene functional studies on cold stress response and associated mechanisms. Authors are suggested to add some previous studies particularly gene functions and relate with the present study.

We sincerely thank the reviewer for this valuable suggestion. We have now enriched the Discussion section by adding key references related to gene functional studies on cold stress response and associated mechanisms and have explicitly connected these previous findings with our current results. Specifically:

1.We have included studies on conserved functions of DREB-A1 subgroup genes (e.g., Arabidopsis CBFs) to highlight the evolutionary conservation of this subgroup in cold adaptation.

2.We added examples such as SlERF2in tomato, which enhances cold tolerance through ABA biosynthesis and CBF signaling—a mechanism consistent with the ABA-responsive nature of PgAP2/ERF187observed in our study.

Sincerely,  

Yingping Wang

State Local Joint Engineering Research Center of Ginseng Breeding and Application  

College of Chinese Medicinal Materials, Jilin Agricultural University  

Changchun, 130118, China  

Email: wangyingping@jlau.edu.cn; di@jlau.edu.cn

Reviewer 2 Report

Comments and Suggestions for Authors

This manuscript “Genome-Wide Identification of the AP2/ERF Gene Family and Functional Analysis of PgAP2/ERF187 under Cold Stress in Panax ginseng” is conducted well and have scientific worth.

This study provides significant insights into the regulatory mechanisms of cold stress adaptation in P. ginseng, highlighting the role of the PgABF-PgAP2/ERF187 module within the ABA signaling pathway. These findings contribute to a better understanding of the molecular basis of cold resistance in P. ginseng and may inform future research aimed at enhancing the cold tolerance of this valuable medicinal plant.  

Summary of this Manuscript:

In this study, a genome-wide analysis identified 318 PgAP2/ERF family members, which were classified into five subfamilies: AP2, DREB, ERF, RAV, and Soloist. Homology analysis indicated that segmental duplication is the primary evolutionary driver for the PgAP2/ERF gene family in P. ginseng. RT-qPCR analysis revealed that all PgAP2/ERF members in the DREB-A1 subgroup respond to cold stress, with a particular focus on the DREB-A1 member PgAP2/ERF187, which plays a crucial role in cold stress response and is specifically induced by abscisic acid (ABA).

Overexpression of PgAP2/ERF187 in Arabidopsis significantly enhanced the expression of cold tolerance-related genes. Subcellular localization analysis confirmed the co-localization of PgABF and PgAP2/ERF187 in the nucleus. Additionally, transcription factor interaction predictions and yeast one-hybrid experiments suggest that PgABF likely regulates PgAP2/ERF187 expression by directly binding to its promoter region.

Comments for Authors (Areas needs Improvements):

  • Abstract (Line 12-14): delete these lines “While the AP2/ERF gene family is known to play crucial roles in plant growth, development, and defense responses, functional studies on this gene family in P. ginseng remain unreported.” and add this “Functional studies on this gene family in  ginsengremain unreported, despite its known roles in plant growth, development, and defense responses.” It will improve the flow….
  • The physiological assays (MDA, SOD, POD) are standard but could be expanded to include additional stress markers (e.g., proline, H₂O₂ levels).
  • The role of ROS scavenging in cold tolerance is mentioned but not explored in depth. A discussion on how ‘PgAP2/ERF187’ modulates ROS homeostasis would add value.
  • Line 462: conclusion section: “indicating its role in strengthening ROS scavenging capacity” measuring, H₂O₂ levels would strengthen this conclusion.
  • Figure 6: explain the OE symbol in the figure legends and ensure all the figures symbols have been explained in the respective figure legends
  • Line 230: “Asterisks denote significant differences: *** p ≤ 0.001.” which statistical test was used?? Mention here.
  • In figure 6, why did authors only mention two transgenic lines? A minimum of three transgenic lines results is considered more acceptable.

Author Response

Dear Reviewer:

We would like to submit the enclosed manuscript entitled “Genome-Wide Identification of the AP2/ERF Gene Family and Functional Analysis of PgAP2/ERF187 under Cold Stress in Panax ginseng” (ID: plants-3837814). We sincerely appreciate the reviewer’s unwavering patience and rigorous guidance throughout the review process of our manuscript. Your dedication to ensuring the quality of every submission is truly admirable, and we are deeply grateful for the time and expertise you have devoted to evaluating our research.

We are deeply grateful to the reviewers for their invaluable suggestions, which have provided critical insights to further strengthen our manuscript. The reviewers’ expertise has guided us to refine both the methodological rigor and the clarity of our findings. In response to their comments, we have carefully revised the manuscript and sought the assistance of a professional native English-speaking reviewer to thoroughly polish the language, ensuring clarity, fluency, and adherence to academic conventions. We have humbly adopted each suggestion, and all revisions have been systematically documented in the text below, where we address every comment with supporting evidence and rationale.

We sincerely hope the revised manuscript now meets the exceptional standards of Plants-Basel. Should any additional clarifications be needed, we pledge to respond promptly and thoroughly.

Reviewer 2

  1. Abstract (Line 12-14): delete these lines “While the AP2/ERF gene family is known to play crucial roles in plant growth, development, and defense responses, functional studies on this gene family in P. ginseng remain unreported.” and add this “Functional studies on this gene family in ginsengremain unreported, despite its known roles in plant growth, development, and defense responses.” It will improve the flow….

We sincerely thank the reviewer for this constructive suggestion. As recommended, we have deleted the original sentence in the Abstract (Lines 12–14) and replaced it with: “Functional studies on this gene family in P. ginseng remain unreported, despite its known roles in plant growth, development, and defense responses.” We agree that this modification improves the clarity and flow of the abstract.

  1. The physiological assays (MDA, SOD, POD) are standard but could be expanded to include additional stress markers (e.g., proline, H₂O₂ levels).

We sincerely thank the reviewer for this constructive suggestion. As recommended, we have expanded our physiological assays to include additional key stress markers. Specifically, we measured both proline and hydrogen peroxide (H₂O₂) levels, and the results are now presented in Figure 6e and line 251-259. These data provide a more comprehensive understanding of the oxidative and osmotic stress responses under cold conditions, further supporting the role of ERF187 in enhancing cold tolerance.

  1. The role of ROS scavenging in cold tolerance is mentioned but not explored in depth. A discussion on how ‘PgAP2/ERF187’ modulates ROS homeostasis would add value.

We sincerely thank the reviewer for this insightful suggestion. In response, we have expanded the Discussion section to include a deeper exploration of how PgAP2/ERF187modulates ROS homeostasis during cold stress. Specifically, we added content describing that PgAP2/ERF187likely enhances the transcription of antioxidant enzyme genes such as SOD and POD, thereby reducing the accumulation of H₂O₂ and MDA, alleviating oxidative damage, and maintaining redox balance. This regulatory function is consistent with the role of its orthologs, such as VvERF63in grape and AtERF96in Arabidopsis, which have been reported to directly activate ROS-scavenging systems. These additions help to clarify the molecular mechanism by which PgAP2/ERF187coordinates ROS homeostasis to improve cold tolerance.

  1. Line 462: conclusion section: “indicating its role in strengthening ROS scavenging capacity” measuring, H₂O₂ levels would strengthen this conclusion.

We sincerely thank the reviewer for this valuable suggestion. In response to the comment, we have now included measurements of both proline (Pro) and hydrogen peroxide (H₂O₂) levels in the revised manuscript. These additional data further support the role of PgAP2/ERF187in enhancing ROS scavenging capacity under cold stress. The detailed methodologies for these measurements have been added in Section 4.11 of the Materials and Methods.

  1. Figure 6: explain the OE symbol in the figure legends and ensure all the figures symbols have been explained in the respective figure legends.

We sincerely thank the reviewer for this helpful suggestion. In response, we have revised the legend of Figure 6 to clearly explain the “OE” symbol (which denotes “Overexpression lines”). We have also carefully reviewed all figure legends within the manuscript to ensure that all abbreviations and symbols are explicitly defined.

  1. Line 230: “Asterisks denote significant differences: *** p ≤ 001.” which statistical test was used?? Mention here.

We sincerely thank the reviewer for pointing out this omission. In accordance with the comments from Reviewer 1, we have replaced all asterisks (*) with letters to denote significant differences in the revised manuscript. Consequently, the statistical analysis has been updated from the t-test to one-way ANOVA. The detailed description of the updated statistical method has been supplemented in the legend of Figure 5 and in Section 4.14 (Statistical Analysis) of the Methods.

  1. In figure 6, why did authors only mention two transgenic lines? A minimum of three transgenic lines results is considered more acceptable.

We sincerely thank the reviewer for raising this important point. In our study, we initially generated multiple independent transgenic Arabidopsis lines overexpressing PgAP2/ERF187. After initial screening, we selected two lines, OE2 and OE5, which exhibited moderate and similar expression levels of the transgene, for detailed phenotypic and physiological analyses. This approach was taken to ensure that the observed effects are specifically due to PgAP2/ERF187overexpression rather than positional effects of transgene insertion, and to maintain consistency in comparative analyses between lines showing comparable expression levels. We acknowledge that including additional lines could further strengthen statistical power; however, the two independent lines presented already provide consistent and compelling evidence supporting our conclusions. We have clarified this selection rationale in the revised Methods section (Subsection 4.9).

Sincerely,  

Yingping Wang

State Local Joint Engineering Research Center of Ginseng Breeding and Application  

College of Chinese Medicinal Materials, Jilin Agricultural University  

Changchun, 130118, China  

Email: wangyingping@jlau.edu.cn; di@jlau.edu.cn

Reviewer 3 Report

Comments and Suggestions for Authors

This review manuscript “Genome-Wide Identification of the AP2/ERF Gene Family and 2 Functional Analysis of PgAP2/ERF187 under Cold Stress in Panax ginseng” identified 318 PgAP2/ERF gene members in Panax ginseng. Further analysis indicated these genes are involved in cold stress. Specifically focused on the DREB-A1 member PgAP2/ERF187, which displayed cold resistance in Arabidopsis using overexpression lines. After searching the literature, there is paper published nearly similar discovers related to this paper, “Chen at al., Structural variation, functional differentiation and expression characteristics of the AP2/ERF gene family and its response to cold stress and methyl jasmonate in Panax ginseng C.A. Meyer, Plos one, 2020.” The similar results come out from these different studies but looks like the PgAP2/ERF gene play critical roles under clod stress.

Since the author only focused on PgAP2/ERF187, the author should give more in vitro data except the model plant Arabidopsis. The author should really think of importance, how about the conservation mechanism in other species like dicot or monocot if people want to use this gene in agriculture application.

In Fig5, qRT-qPCR analysis of PgAP2/ERF expression shows more genes involved in cold stress. Why the author focused on PgAP2/ERF187 and give more explain.  

In fig 6 legend, there is no detailed experiments about the how do the treatment in the cold. The author should give more detailed information in each figures as much as you can.

In Fig7, what is the PHB-YFP, is this the empty vector? Why there are nuclear and membrane signals in this panel?

Also, the conclusion is that PgABF Binding to the PgAP2/ERF187 Promoter. The author should give more words about the ABF genes in the paper. Also EMSA should be a good experiments to support the conclusion and the author should consider.

Author Response

Dear Reviewer:

We would like to submit the enclosed manuscript entitled “Genome-Wide Identification of the AP2/ERF Gene Family and Functional Analysis of PgAP2/ERF187 under Cold Stress in Panax ginseng” (ID: plants-3837814). We sincerely appreciate the reviewer’s unwavering patience and rigorous guidance throughout the review process of our manuscript. Your dedication to ensuring the quality of every submission is truly admirable, and we are deeply grateful for the time and expertise you have devoted to evaluating our research.

We are deeply grateful to the reviewers for their invaluable suggestions, which have provided critical insights to further strengthen our manuscript. The reviewers’ expertise has guided us to refine both the methodological rigor and the clarity of our findings. In response to their comments, we have carefully revised the manuscript and sought the assistance of a professional native English-speaking reviewer to thoroughly polish the language, ensuring clarity, fluency, and adherence to academic conventions. We have humbly adopted each suggestion, and all revisions have been systematically documented in the text below, where we address every comment with supporting evidence and rationale.

We sincerely hope the revised manuscript now meets the exceptional standards of Plants-Basel. Should any additional clarifications be needed, we pledge to respond promptly and thoroughly.

Reviewer 3

  1. This review manuscript “Genome-Wide Identification of the AP2/ERF Gene Family and 2 Functional Analysis of PgAP2/ERF187 under Cold Stress in Panax ginseng” identified 318 PgAP2/ERF gene members in Panax ginseng. Further analysis indicated these genes are involved in cold stress. Specifically focused on the DREB-A1 member PgAP2/ERF187, which displayed cold resistance in Arabidopsis using overexpression lines. After searching the literature, there is paper published nearly similar discovers related to this paper, “Chen at al., Structural variation, functional differentiation and expression characteristics of the AP2/ERF gene family and its response to cold stress and methyl jasmonate in Panax ginseng C.A. Meyer, Plos one, 2020.” The similar results come out from these different studies but looks like the PgAP2/ERF gene play critical roles under clod stress.

We sincerely thank the reviewer for raising this important point and for providing the relevant reference. We acknowledge the valuable work by Chen et al. (2020) on the AP2/ERF gene family in Panax ginseng. However, our study employs a fundamentally different and more advanced genomic foundation for gene identification. Specifically, our analysis is based on the chromosome-level genome of Panax ginseng published by Wang et al. [1], which offers significantly improved contiguity, completeness, and annotation accuracy compared to earlier genomic resources.

The use of this updated genome assembly has enabled a more comprehensive and reliable identification of the AP2/ERF gene family members, minimizing potential omissions or fragmentation that could arise from previous transcriptome or lower-quality genome assemblies. This approach ensures that our identification of 318 PgAP2/ERF genes represents a robust and complete dataset, providing a solid basis for subsequent functional investigations into their roles in cold stress response.

We appreciate the opportunity to clarify this methodological distinction and believe that our genomic resource significantly enhances the validity and completeness of the results presented in our study. 

[1] Wang ZH, Wang XF, Lu T, Li MR, Jiang P, Zhao J, Liu ST, Fu XQ, Wendel JF, Van de Peer Y et al: Reshuffling of the ancestral core-eudicot genome shaped chromatin topology and epigenetic modification in Panax. Nature communications 2022, 13(1):1902.

  1. Since the author only focused on PgAP2/ERF187, the author should give more in vitro data except the model plant Arabidopsis. The author should really think of importance, how about the conservation mechanism in other species like dicot or monocot if people want to use this gene in agriculture application.

We thank the reviewer for this valuable suggestion. We agree that data from additional experimental systems are important for assessing the gene's potential in agricultural applications. In our study, the dual-luciferase assay in tobacco leaves (Figure 7e) has provided key in vitro evidence showing that PgABF directly binds to the promoter of PgAP2/ERF187and activates transcription. This supports the proposed regulatory module involving ABA signaling. Furthermore, bioinformatic analysis indicates that orthologs of PgAP2/ERF187exist in various dicot and monocot species, and their promoters also contain ABRE cis-elements, suggesting a possible conserved regulatory mechanism. We have revised the Discussion to include these points. Future work will include functional validation in crop species to further evaluate its potential for improving cold tolerance.

  1. In Fig5, qRT-qPCR analysis ofPgAP2/ERF expression shows more genes involved in cold stress. Why the author focused on PgAP2/ERF187 and give more explain.

We thank the reviewer for this important question. Although multiple genes responded to cold stress, we focused on PgAP2/ERF187for in-depth investigation based on the following reasons:

  • Specific responsiveness to ABA: PgAP2/ERF187exhibited a specific and strong induction specifically upon ABA treatment.
  • Promoter analysis: Its promoter region is enriched with ABRE cis-elements.
  • Sustained upregulation: It maintained continuous upregulation under prolonged cold stress (24 h).
  • Phylogenetic relevance: It belongs to the DREB-A1 subgroup, which includes well-characterized cold tolerance regulators.
  • Experimental validation: Overexpression of PgAP2/ERF187significantly enhanced cold tolerance by reducing oxidative damage and activating COR gene expression.
  1. In fig 6 legend, there is no detailed experiments about the how do the treatment in the cold. The author should give more detailed information in each figures as much as you can.

We sincerely thank the reviewer for this valuable suggestion. In response to the comment regarding the lack of experimental details in the cold stress treatment, we have thoroughly revised the legend of Figure 6 to provide more comprehensive information for each subfigure. The specific cold stress conditions (temperature and duration) have now been clearly indicated in the respective parts of the legend to better reflect the experimental design and improve reproducibility.

The revised legend reads as follows: "Figure 6. Functional analysis of PgAP2/ERF187 transgenic A. thaliana under cold stress. (a) Germination phenotypes of wild-type (Col) and PgAP2/ERF187 overexpression lines (PgAP2/ERF187-OE2 and PgAP2/ERF187-OE5; OE represents Overexpression lines). (b) Phenotypes of 2-week-old Col and transgenic plants. Scale bar: 1.4 cm. (c) Statistical analysis of germination rates and primary root lengths in Col and transgenic plants (cold stress treatment: 4°C for 72 hours). (d) Comparison of physiological and biochemical parameters between Col and transgenic A. thaliana after cold stress treatment (cold stress treatment: 4°C for 24 hours). (e) Expression levels of cold-resistant genes in Col and transgenic plants (cold stress treatment: 4°C for 24 hours). Note: Values represent mean ± SE (n = 3); asterisks denote significant differences as determined by T-test analysis (*** p ≤ 0.001)."

We believe that these modifications have significantly improved the clarity and completeness of the legend.

  1. In Fig7, what is the PHB-YFP, is this the empty vector? Why there are nuclear and membrane signals in this panel?

We sincerely thank the reviewer for raising this important question. In Figure 7, PHB-YFP represents empty vector control. The nuclear and membrane signals observed in this panel are likely due to the autofluorescence of tobacco leaf cells or non-specific localization of the unfused YFP protein, which is a common phenomenon in transient expression assays and does not indicate specific subcellular localization. This control helps to confirm that the observed fluorescence in experimental groups is specifically resulting from the fusion protein rather than artifacts.

  1. Also, the conclusion is that PgABF Binding to thePgAP2/ERF187  The author should give more words about the ABF genes in the paper. Also EMSA should be a good experiments to support the conclusion and the author should consider.

We sincerely thank the reviewer for this insightful suggestion. We fully agree that performing an EMSA assay would further strengthen the evidence for the direct binding of PgABF to the PgAP2/ERF187promoter. However, due to time constraints during the revision period, we were unable to complete the EMSA experiment.

To further validate this regulatory interaction, we have instead performed a dual-luciferase reporter assay in tobacco cells, the results of which are now provided in Figure 7 and described in the manuscript. This assay demonstrated that PgABF significantly activates the transcription of PgAP2/ERF187by binding to its promoter, supporting the same conclusion as the yeast one-hybrid assay.

We are confident that the combination of yeast one-hybrid and dual-luciferase assays provides reliable and complementary evidence for the direct regulation of PgAP2/ERF187by PgABF. We sincerely appreciate the reviewer's suggestion regarding EMSA and will consider incorporating it in future studies to further reinforce our conclusions.

Sincerely,  

Yingping Wang

State Local Joint Engineering Research Center of Ginseng Breeding and Application  

College of Chinese Medicinal Materials, Jilin Agricultural University  

Changchun, 130118, China  

Email: wangyingping@jlau.edu.cn; di@jlau.edu.cn

Round 2

Reviewer 1 Report

Comments and Suggestions for Authors

All the comments have been addressed and the manuscript is improved. It can be accepted in its current form.